# Legionella in Primary School Hot Water Systems from Two Municipalities in the Danish Capital Region

**DOI:** 10.3390/microorganisms12102074

**Published:** 2024-10-16

**Authors:** Niss Skov Nielsen, Peter Fojan, Rasmus Lund Jensen, Haseebullah Wahedi, Alireza Afshari

**Affiliations:** 1Division of Building Technology, Management and Indoor Environment, Danish Building Research Institute (Build), Aalborg University, A.C. Meyers Vaenge 15, 2450 Copenhagen, Denmark; rlje@build.aau.dk (R.L.J.); haw@build.aau.dk (H.W.); aaf@build.aau.dk (A.A.); 2Centre for Health Research, Zealand University Hospital, Strandboulevarden 64, 4800 Nykøbing Falster, Denmark; 3Department of Materials and Production, Aalborg University, Fibigerstræde 14-16, 9220 Aalborg, Denmark; fp@mp.aau.dk

**Keywords:** colonisation rate, Legionnaires’ disease, preventive procedures, primary schools, test procedures, water temperature

## Abstract

Legionella contamination in public water systems poses significant health risks, particularly in schools where vulnerable populations, including children, regularly use these facilities. This study investigates the presence of Legionella in the hot water systems from 49 primary schools across two municipalities in the Danish capital region. Water samples were collected from taps in each school, and both first-flush and stabile temperature samples were analysed for Legionella contents. The findings revealed that 97% of schools in Municipality 1 and 100% in Municipality 2 had Legionella in their hot water systems. The content of Legionella colonies was significantly higher in schools in Municipality 1, which was probably because of overall lower water temperatures. At stabile temperatures, 76% and 50% of the schools in the two municipalities exceeded the European Union’s recommended limit of 1000 CFU/L. Stabile peripheral water temperatures were achieved after 3 min. Tap water temperatures above 54 °C and central tank temperatures above 59 °C were associated with Legionella contents below 1000 CFU/L. This study highlights the need for more stringent Legionella control procedures in schools, including higher water temperatures and refining Legionella reducing interventions with the addition of regular flow and draining procedures.

## 1. Introduction

*Legionella* (L.) pneumophila is a waterborne pathogen that can cause severe life-threatening pneumonia called Legionnaires’ disease (LD) [1]. Denmark is one of the countries in Europe with the highest reported incidence of LD [2]. The risk of contracting LD and severe pneumonia increases with age, while younger and healthy people more often have courses with symptoms reminiscent of milder respiratory infections called Pontiac fever [1,3]. Even so, *Legionella* infection is a health risk among vulnerable persons in all age groups.

The quality of water systems in schools and their operation are of particular relevance for limiting the risk of acquiring a *Legionella* infection among children but also among other age groups, because school facilities are often used by the elderly and by other vulnerable groups during off-hours. Danish municipalities regulate and control such risks in schools [4].

The Danish National Serum Institute (SSI) has until recently recommended a maximum content of 1000 *Legionella* colony-forming units per litre (CFU/L) (*L. pneumophilia* serogroup 1) and up to 10,000 CFU/L (serogroups 2–14) in hot water systems [5,6]. Municipalities have used these guidelines to regulate *Legionella* contents in Danish hot water systems, even though the recommendations are not official governmental limits [7].

For almost a decade, the WHO [1] has recommended national limits of 1000 *Legionella* (CFU/L) for all serogroups in hot water systems. The European Union (EU) has recently followed this recommendation with their Drinking Water Directive and introduced a similar limit for *Legionella* levels in hot water systems in public buildings and in those used by vulnerable people [8]. A maximum content of 1000 CFU/L for *Legionella* in hot water systems will thus be introduced in public buildings in EU member states shortly.

A recent study of Danish care homes showed that approximately 80% of the institutions’ hot water systems were colonised with *Legionella* and that the mean contents were 35,784 CFU/L in A-samples (first-flush), with 7962 CFU/L in B-samples (at stabile water temperature). The maximum contents of *Legionella* in A-samples were 600,000 CFU/L and up to 134,000 CFU/L in B-samples in the hot water systems from the investigated care homes [9].

Furthermore, 40% of the care homes exceeded 1000 CFU/L based on the B-samples. This was up to 60% when A-samples were included. Finally, the results showed that 15% of the care homes exceeded 10,000 CFU/L based on cultured B-samples, and this was 30% when A-samples were included [9].

Many Danish primary schools were built in the 1960s–1970s, when the average water use per person was much higher than today [10,11]. Thus, many of these older school water systems have much higher water capacity than needed [12]. This can promote favourable conditions for *Legionella* biofilm formation and growth [4,7,13].

Moreover, few pupils, especially in the older classes, currently use the bathing facilities in primary schools [14]. This contributes to lower water use during regular school hours and increases the risk of *Legionella* growth and biofilm contamination in the schools’ hot water systems [13]. However, many school facilities are currently used by other age groups during off-hours [15]. This might partially compensate for the lower water usage during the daytime in primary schools but also pose a risk for vulnerable individuals due to potential Legionella exposure.

High water temperatures are effective at controlling *Legionella* growth. Of particular interest, water temperatures consistently held over 55 °C seem to be effective [1,16]. The effects of thermal shock (e.g., increasing the water temperature once a week to 70 °C) combined with regular decalcification of the central boilers and long-distance CTS management (Central Condition, Control, and Management, e.g., with a building management system (BMS) to control water temperature from a long distance) resulted in a reduction of 70% of the *Legionella* levels in Danish care homes after three months [7]. Many Danish municipalities have since introduced such *Legionella*-reducing procedures to control *Legionella* growth in public institutions. The operational optimisation intervention results lack evidence of long-term effects [16]. Such procedures may cause a selection of thermally tolerant bacterial strains if the intermittent thermal shocks are combined with a generally low water temperature for the remainder of the time. Such a selection is of concern because thermal-tolerant strains may more easily harm human health than other strains [17]. The low use of some taps connected to hot water systems might cause peripheral accumulation of *Legionella* strains near these taps. From backflow, these may interact with the central parts of the hot water system if the taps are not used regularly and not flushed in connection with the heath shock procedures [18].

First-flush samples from water taps (A-samples) are typically used to measure the peripheral levels of *Legionella* in taps. Flush samples (B-samples) at stabile temperatures are usually used to measure *Legionella* levels in the central parts of a water system [5]. Both measurements are critical but have a significant risk of bias across studies depending on how the different hot water systems are constructed and used, as well as which sampling procedures are used. Some authorities have recommended using B-samples after 1 or 2 min of flow at half the maximum water speed to ensure an approximately comparable test across water systems [19]. Others have recommended waiting for the stable maximum hot water temperature before taking B-samples to take into account construction differences between the water systems; see ref. [20]. The latter technique might cause a large part of the *Legionella* content to be washed out if it takes several minutes before a stable water temperature is achieved [21]. More precise and standardised procedures seem necessary to ensure reliable and comparable results across hot water systems, especially when a maximum content of 1000 CFU/L will soon be the limit for *Legionella* levels in hot water systems in EU countries [8].

The first aim of this study was to determine the presence of culturable *Legionella* in the hot water systems of the primary schools in two Danish municipalities. The second aim was to analyse differences in *Legionella* concentrations and water temperatures in water samples from schools in the two municipalities. Third, we calculated the relationship between the *Legionella* concentration (CFU/L) in B-samples and water temperature from primary schools in the two municipalities. Fourth, we analysed which water temperatures in the primary schools are correlated with the EU Legionella limit of 1000 CFU/L.

Finally, we analysed the *Legionella* levels in water samples collected over an extended period to investigate the development of temperature and leaching of *Legionella* bacteria over time, with the aim of improving test procedures for Legionella in school water systems.

## 2. Methods

Forty-nine primary schools were randomly selected from two municipalities in the Danish capital region to investigate the presence of Legionella in their hot water systems. All the schools used untreated groundwater and did not treat the water within their premises.

### 2.1. Water Samples

Two samples were collected from the schools’ hot water systems in June and July 2023. Additional samples were collected from two schools in August 2023 to investigate the stability of *Legionella* levels over time. Two samples were collected at each school. One group of the two samples was collected as “first-flush samples” (A-samples), and the other was collected at stable temperature (B-samples). The B-samples were collected following the Danish Serum Institute’s (SSI) recommendations, which should be collected after 2 min of flushing from the taps [5]. Hot water temperatures were measured using alcohol-based thermometers after collecting the 2-min samples.

In seven randomly selected schools, additional samples and temperature measurements were taken to investigate whether 2 min is the ideal time to achieve stable water temperatures from larger water systems in schools. In the seven schools (three and four schools from the two municipalities), four water samples were collected at intervals of 0, 2, 4, and 8 min of flushing, and the water temperature was measured after 2, 3, 4, 5, and 8 min of flushing. These samples were used to investigate the relationship between water temperature and *Legionella* levels over time.

All the water samples were collected from taps in the schools by representatives from Build (Aalborg University) following the recommended criteria of two samples from taps or showers farthest from the central boilers in the hot water system. Filters and water/air mixers were initially removed from the taps before the samples were collected. Then, 1000 mL samples were tapped directly into bottles. These samples were stored at temperatures between 6 and 18 °C until they were tested in the laboratory [5,6]. To investigate whether single taps function as “blind ends” (taps which are not or seldom used) in the water systems, the samples were collected using criteria for the rare use of a distant tap. All the water samples were then transported to accredited firms for testing using the culture method.

### 2.2. Analysis Based on ISO 11731

A portion of the water samples was delivered to the Danish National Serum Institute (SSI) in Copenhagen, Denmark, for culturing. The remaining samples were delivered to Eurofins A/S in Ishoej for similar processing. These samples were transported every evening to their laboratory in Vejen, Jutland, Denmark, for culturing and analysis. The samples were processed within 24 h following the international standards [22]. From the 1000 mL samples, 2 × 500 µL were plated directly onto two GVPC (Oxoid, Fisher Diagnostics, Basingstoke, UK) agar plates, and 1 L was filtered through a 0.22 µm polyethersulfone membrane filter (Micro Funnel Plus, Pall Life Science, Portsmouth, UK). Each filter was then vortexed with glass beads for 4 min with 10 mL of the sample. From each filtrate, 2 × 100 µL and 2 × 500 µL were seeded to GVPC agar plates. The plates were incubated at 36 °C in plastic bags for 7 days. The plates were inspected after 2 days, and if growth of interfering bacteria was observed, aliquots of the filtrate (kept at 4–10 °C) were subjected to heat and acid treatment. Subsequently, 2 × 100 µL from each treated sample were plated on GVPC agar plates and incubated for seven days. The *Legionella* colonies on each plate were counted, and the highest colony count among the three steps was recorded as the result, expressed as the content of colony-forming units per litre (CFU/L). Direct plating was used for enumeration only when ≥10 colonies were identified on the two plates. The limit of detection by the culture method was 100 CFU/L, which was the limit of quantification for both laboratories.

### 2.3. Serogroup Determination

For the portion tested at SSI, a serogroup study as a sample of which types were present in the water systems of the schools in the two municipalities was carried out. At least five colonies (if present) from each positive sample were analysed using the Oxoid *Legionella* Latex Test (Thermo Fisher Scientific, Waltham, MA, USA) to categorise the Legionella types into categories of *L. pneumophila* serogroup 1, serogroups 2–14, and other *Legionella* species. Representative isolates of serogroup 1 were additionally analysed using the Dresden panel of monoclonal antibodies [23] to identify the virulence-associated epitope recognised by monoclonal antibody 3/1 (mAb 3/1) for further risk assessments. In cases where *Legionella* was not detected (<100 CFU/L), a value of 0 was used in the calculations. The culture method is the only accepted method for *Legionella* risk assessment by the Danish Environmental Protection Agency [24].

### 2.4. Other Data

We also asked the municipalities and representatives from the schools for the following data: use of BMS (long-distance control of the water temperature), use of *Legionella*-reducing procedures (heat shock and decalcification/cleaning of central water tank once a year), and the date of the last decalcification/cleaning of the water system.

Between 29 and 31 of the 49 schools were further able to collect the following data: use of catalytic, ionic changing or other water treating equipment, size of the water tank (L), ages of the water tank and water system, and, finally, the water temperature in the central water tank. Only water temperature data from taps and the central water tanks were used in this study.

### 2.5. Statistical Analysis

We used SPSS version 29 (IBM, Armonk, NY, USA) to analyse the data. Results were considered significant when *p* ≤ 0.05. Some results were at interval scale levels but did not follow a Gaussian distribution pattern. Therefore, independent *t*-tests were used for these analyses. The Wilcoxon signed rank test was used to determine the significance between the medians. Two-sided chi-square tests were used to determine the significance between the school percentages of Legionella between the municipalities.

### 2.6. Legionella Levels and Water Temperatures in Schools from Two Municipalities

*Legionella* contents in the primary schools were compared and analysed between the two municipalities, as shown in Table 1 and Table 2. These tables include the distribution of schools from the two municipalities with CFU/L levels above 0 (above 10 CFU/L as the lowest detectable limit), above 1000 CFU/L, and above 10,000 CFU/L.

Differences in mean and median contents between A-samples and B-samples were analysed, as shown in Table 2. Furthermore, the distributions of contents between 100–10,000 CFU/L and above 10,000 CFU/L were analysed across the two municipalities. This was done to investigate whether the distributions in the two municipalities were uniform.

This is further shown in Table 3, where the mean water temperature and mean and median counts of *Legionella* from the B-samples across the two municipalities are compared. The schools are divided into schools with mean water temperatures below 50 °C and on/above 50 °C.

Figure 1 shows a scatterplot and the statistical relationship between tap water temperatures measured after 2 min and the number of *Legionella* counts in B-samples (2 min) from all schools in the two municipalities. Furthermore, a regression model is shown in Figure 1 to illustrate the association between water temperatures and *Legionella* contents.

To illustrate whether 2 min is the ideal time of water flow to achieve stable water temperature in water systems, we prepared Figure 2 as a scatterplot to show the relationship between tap water temperatures measured after 3 min and *Legionella* counts (CFU/L) in B-samples after 2 min.

Hot water temperatures from the central water tanks were obtained from only 32 of the 49 investigated schools. The *Legionella* contents in the B-samples from these schools, along with the central water tank temperature, are illustrated in Figure 3.

The relationship between tap water temperatures and the duration of tap water flushing is shown in Figure 4, which is based on data from seven randomly selected schools from the two municipalities. Water temperatures were measured at 2, 3, 4, 5, and 8 min during the water flushing.

Figure 5 shows the relationship between *Legionella* contents (CFU/L) and duration of tap water flushing based on data from six of seven randomly selected schools. One of the schools had no detectable *Legionella* in the tap water, and data from this school are therefore not included in Figure 5.

## 3. Results

The measurements of the *Legionella* levels in the primary school water systems from the two municipalities revealed that almost all the schools were colonised with *Legionella* (97% in Municipality 1 and 100% in Municipality 2; Table 1). According to the A-samples and B-samples, a significantly higher proportion of the schools contained 10,000 CFU/L or more in Municipality 1 (63%) compared to Municipality 2 (13%) (Table 1).

High *Legionella* contents were observed in some samples from Municipality 1. Up to 100,000 CFU/L was observed in A-samples and B-samples (Table 2). The mean content of *Legionella* was 31,268 CFU/L in the A-samples from the schools with *Legionella.* The results also showed that the content of CFU/L in B-samples (17,300 CFU/L) was approximately half that in the A-samples, which represents a considerable but not a significant difference between the mean contents in A-samples compared to B-samples (*p* = 0.064). The median counts were significantly lower than the mean counts in the A-samples and B-samples in Municipality 1.

The proportions of schools in Municipality 1 that exceeded 10,000 CFU/L were 51% for A-samples and 32% for B-samples, representing a significant difference (*p* = 0.046). In addition, 78% and 76% of the schools exceeded 1000 CFU/L in A-samples and B-samples, respectively (sum the two last column proportions in Table 2).

In Municipality 2, we observed the highest number of *Legionella* units at 21,000 CFU/L in an A-sample and 3400 CFU/L in one of the B-samples (Table 2). The mean content in A-samples from schools with *Legionella* was 4120 CFU/L in Municipality 2. The mean content of *Legionella* in B-samples (1850 CFU/L) was also approximately half the mean content found in A-samples (4120 CFU/L), but this difference was not significant (*p* = 0.528). The mean counts in the A-samples (4120 CFU/L) were higher than the median counts (1200 CFU/L) but not significantly higher (*p* = 0.130). In B-samples, the mean and median counts were almost similar (1850 vs. 2150 CFU/L).

The proportion of schools that exceeded 10,000 CFU/L was 13% for A-samples, but none of the B-samples in Municipality 2 exceeded 10,000 CFU/L. Furthermore, 50% of the A-samples and B-samples exceeded 1000 CFU/L in Municipality 2.

The comparison between the two municipalities showed that the mean contents in A-samples (*p* = 0.005 and 0.004) and B-samples (*p* = 0.041 and 0.015) were significantly higher in Municipality 1 than in Municipality 2, respectively. Based on the proportions of schools with more than 0, 1000–10,000, and more than 10,000 CFU/L, the distribution between A-samples and B-samples was similar in the two municipalities (Table 2).

Table 3 shows that the mean water temperature below 50 °C in schools in Municipality 1 (41.5 °C) was more than 4 °C lower than in schools from Municipality 2 (45.7 °C). The corresponding mean content of *Legionella* in these schools was significantly higher (*p* = 0.011) in Municipality 1 (21,694 CFU/L) than in Municipality 2 (1577 CFU/L). For schools with a water temperature ≥50 °C, the mean water temperature was almost the same in the two municipalities (52.9 vs. 53.0 °C). However, the mean content of *Legionella* in the schools from Municipality 1 was significantly higher (*p* = 0.027) (1950 CFU/L) than the mean content in the schools from Municipality 2 (100 CFU/L).

The regression model In Figure 1 between the temperature (“constant” temperatures measured after 2 min of water flow from the taps) and the *Legionella* count in the water samples showed that the regression line was significant for Municipality 1 (*p* = 0.016; R2 = 0.121), and the regression formula of the line was y = 6.69 × 10^4^–1.18 × 10^3x^. Figure 1 also shows that the maximum water temperature in the tap water achieved after 2 min of flow was 54 °C. Moreover, temperatures of 50 °C or higher were correlated with levels of *Legionella* below 10,000 CFU/L for all schools in the two municipalities. No clear temperature cut-off point was observed for levels below 1000 CFU/L (see Figure 1).

Figure 2 shows that the maximum tap water temperature was up to 57 °C after 3 min of flow. Furthermore, water temperatures above 54 °C were related to a maximum *Legionella* content of 1000 CFU/L (measured after 2 min), and water temperatures above 50 °C were related to a maximum *Legionella* content below 10,000 CFU/L.

Table 3 shows that only central tank temperatures of 60 °C were related to a *Legionella* content of below 1000 CFU/L.

Figure 4 show that a stable water temperature in the tap water was achieved after 3 min of flow from the taps. These results are based on results from seven schools.

Figure 5 show how the *Legionella* content in water samples is related to the number of minutes of water flow. There was a considerable decrease in *Legionella* content between 0 and 2 min of flow. After 2 min flow, the decrease in *Legionella* content was much lower. Based on the results from six schools, the *Legionella* content after 3 min was approximately 14% lower (1170 CFU/L) than the content measured after 2 min of flow (1367 CFU/L).

Colonies from the cultures of the seven schools showed that *L. pneumophila* serogroups 2–14 were identified in the samples from six of the seven schools, while no cultivable Legionella was detected in the water system of the seventh school. Furthermore, *L. aniisa* was identified in two of the schools. No viable *L. pneumophila serogroup 1* was identified in any of the hot water systems of the investigated schools.

## 4. Discussion

Almost all the schools tested in the two municipalities were found to be colonised with *Legionella* in their hot water systems (97% in Municipality 1 and 100% in Municipality 2). In both municipalities, the mean contents of *Legionella* in A-samples were approximately two times higher than those in B-samples.

The mean contents of *Legionella* (CFU/L) in the A-samples and B-samples were significantly higher in Municipality 1 than in Municipality 2. Moreover, significantly more schools in Municipality 1 (63% of the schools) had a *Legionella* content above 10,000 CFU/L than in Municipality 2 (13% of the schools). In addition, 76% of the schools in Municipality 1 had B-samples with *Legionella* concentrations above 1000 CFU/L. The proportion was 50% in Municipality 2.

Additionally, the water temperatures in schools from Municipality 1 were generally lower than in schools from Municipality 2. This was especially the case among the schools with water temperatures below 50 °C, where the mean water temperature in Municipality 1 was approximately 4 °C lower than in schools from Municipality 2.

A regression line was calculated between the tap water temperatures and contents of *Legionella* in schools from the two municipalities. Tap water temperatures of 55 °C or higher measured after 3 min of flow and a central water tank temperature of 60 °C were threshold values, and none of these samples exceeded 1000 CFU/L. Finally, the cultivation of water samples from a selection of schools from the two municipalities showed that only low virulent *Legionella* serogroups were present in the water systems.

Bias is typically related to investigators using different collection procedures. Only two Build (Aalborg University) investigators collected all the water samples after the described procedures (see the Method section) to reduce such bias. To further reduce such bias, water samples from the first four schools in this study were collected by both investigators.

We used two different laboratories to cultivate the water samples in this investigation. Because one of the laboratories is far from the geographic test area, this could have introduced bias due to the time differences between sample collection and cultivation. To avoid such potential errors, both laboratories cultivated the samples in this study the day after they were delivered. Another potential source of errors could be differences in the cultivation procedures between the laboratories. Although both laboratories used the same ISO-certified methods (ISO Standard 11731) [22] and used the same limits of detection and of quantification, it is impossible to eliminate minor variations in the procedures across laboratories, which might have caused differences in the cultivation results. Overall, the *Legionella* measurements from the two laboratories were of the same order of magnitude, which indicates that errors based on different procedures between the laboratories were minor.

Potential differences might also be related to the calculations related to water temperature and the number of *Legionella* colonies over time (Figure 4 and Figure 5), which are based on results from six and seven schools. However, the same trends in water temperatures and *Legionella* contents were observed in all the six and seven schools over time, which indicates that the mean values may be different from the results from all 49 schools. But for the overall trends regarding the increase in water temperature over time (Figure 4) and the decrease in *Legionella* contents over time (Figure 5), the results would likely be the same as if data from all 49 schools were included.

Because the sample on the cultivation of the different serogroups present in the water systems was based on the same seven schools, more virulent types could have been present in other schools, resulting in limitations in our results. As the new EU limit for Legionella content in water systems is independent of serotype, further analyses of serotypes were out of the scope of this study.

We used results from both municipalities regarding regression and relationships between water temperatures and *Legionella* contents in the schools, even though the results showed that the contents of *Legionella* were significantly higher in Municipality 1 (Figure 1, Figure 2 and Figure 3) and that water temperatures were generally lower in schools from Municipality 1 than in Municipality 2 (Table 3). This might result in errors in the regression line shown in Figure 1, even though the slope of this line is very near the slope of the regression line observed in a care home study [9]. However, this is likely not the case for schools with high water temperatures, which were used in this study to show water temperature limits for *Legionella* contents of 1000 CFU/L. This was because only minor water temperature differences were observed between the schools from the two municipalities for water temperatures above 50 °C (Table 3).

Most of our results showed that schools with high water temperatures showed a correlation between high water temperatures and low contents of *Legionella*. In contrast, the results for schools with lower water temperatures showed divergent trends between water temperatures and *Legionella* contents in the water systems. This is one of the main reasons a Gaussian distribution was not observed between the results, which means that non-parametric statistics were primarily used in this study. Other studies have also shown a non-Gaussian distribution of *Legionella* contents related to water temperature for B-sample testing [9].

First-flush samples (A-samples) are expected to yield a higher level of colonies than flush samples collected at constant temperatures (B-samples), as reported by Hirsh et al. [19]. In this study, the A-samples had two times the content of the B-samples, which is consistent with findings from other *Legionella* studies [9].

The significantly higher contents of *Legionella* in both A-samples and B-samples from schools in Municipality 1 than in Municipality 2 was most likely primarily caused by the 4 °C lower mean water temperature in the schools from Municipality 1. This was specifically observed in schools with water temperatures below 50 °C (Table 3). This may also partially explain why a much higher proportion of schools in Municipality 1 than those in Municipality 2 had a *Legionella* content exceeding 10,000 CFU/L.

The much higher contents of *Legionella* in A-samples, particularly in some schools from Municipality 1 (up to 100,000 CFU/L) compared to the maximum contents in schools from Municipality 2 (up to 31,268 CFU/L), further indicated this temperature difference between the two municipalities. Another possibility is that these schools have taps that are rarely used. This could result in “technical blind ends” in some of the water systems, which are not affected by the heat shock procedures, as circulation or regularly draining water from taps is not a mandatory part of the Legionella-reducing procedures in most schools. This might result in lower water temperatures in peripheral tubes in the water systems and accumulations of Legionella colonies that interact with the water system’s central part after the thermal shock procedure, allowing for a quick reestablishment of Legionella in the central part of the water system. Furthermore, a short increase in water temperatures during the heat shock is not sufficient to sterilize a water system, as part of the colonies might change into a VBNC phase, which subsequently can be transformed into active bacteria again when conditions are favourable again [25,26].

In addition, such procedures could also lead to a selection and presence of more thermos-tolerant Legionella types, of which *L. pneumophila* serogroup 1 is a well-known example, which is particularly contagious [27].

We asked for water samples from taps that are seldom used in schools to illustrate such potential accumulations. Equivalent high mean counts of colonies in B-samples in some schools support that this might be the case for some schools, given that the water temperatures were not particularly low in these schools in Municipality 1. The general significant differences between the mean and median scores for the B-samples in Municipality 1 show that this might only be the case for a few of the schools.

Another possible explanation for the high contents of *Legionella* in some water systems is that effects from *Legionella*-reducing procedures, such as thermal shock, may stagnate after some time. Studies have shown stagnation in the effects after 6 months [28]. Other studies have shown that a selection against more thermally resistant species of *Legionella* occurs after some time with thermal shock procedures [17]. High calcium contents in the water and low water flow from a part of the water system may promote biofilm growth, which to some degree protects and promotes further growth of *Legionella*. This could be a further reason for the high levels of *Legionella* in some of the water systems [13,29].

The recommended procedure for collecting water samples at stable temperature has so far been to wait for 1 min for stable hot water temperature but not wait for more than 2 min before collecting B-samples [5,30,31]. As the cultivation method is very time-sensitive to the leaching of colonies [30], this procedure will likely result in differences in the number of colonies in B-samples based on different time periods used for collecting samples. Consequently, we collected the water samples after 2 min to avoid such differences. As shown in Figure 2, Figure 4, and Figure 5, this procedure did not result in stable water temperatures in all the water systems. Instead, the results indicated that the correct procedure should have been to wait 3 min so that the water temperature was stable. The question is whether the samples should have been collected after 3 min. As shown in Figure 5, the number of colonies did not decrease considerably between 2 and 3 min (14%). Furthermore, upon comparing the number of colonies in Figure 2 with Figure 1, we found that the number of colonies measured after 2 min related to the tap water temperature measured after 3 min was the maximum at 1000 CFU/L in water systems with water temperatures above 55 °C. A procedure to avoid dramatic leaching of *Legionella* colonies by collecting tap water samples after 2 min and then waiting until 3 min before measuring the water temperature might be a better procedure for investigating large central water systems in schools. An alternative procedure could be to collect the tap water samples and measure the water temperatures after 3 min and then consider a reduction of 14% in *Legionella* content compared to measurements collected after 2 min.

Although only less virulent types of *L. pneumophila* and *L. anisa* were detected in the schools from the two municipalities in this study, the high levels in some samples and the large proportions of schools with *Legionella* in hot water systems pose a risk to vulnerable users of the facilities in these schools. A measurement of culturable *Legionella* colonies of up to 100,000 CFU/L is very high compared to the recommended limits of 1000 CFU/L from the EU and even compared to the former recommended limit of 10,000 CFU/L [5], independent of the type of serogroups present [1,5]. No LD cases are known to be associated with the schools in this study, which might be due to the low virulence of the *Legionella* strains found in this study. Alternatively, many school children might have suffered from a milder form of *Legionella* infection called Pontiac fever [32] and have confused this with, for example, a flu infection, as an etiological diagnosis is not always established in such cases [2,4].

This study’s proportion of schools with *Legionella* is uniquely high, as almost all the institutions had *Legionella* colonies in their hot water systems. In other Danish studies, the proportion of care homes that showed *Legionella* was 80% [9], and a study of older apartments showed that 70% showed *Legionella* [31].

Regarding the results from other countries, studies from Italy showed a *Legionella* colonisation rate of 93.7% in hospitals [33,34]. The results from Poland showed colonisation rates between 41 and 74.8% [35,36]. Finally, a study from Turkey showed that 69.2% of hotels were colonised with *Legionella* [37]. Importantly, water systems in many other countries are chlorinated or treated differently than the water in Danish hot water systems. This might explain the higher proportion of institutions in Denmark infected with *Legionella* compared to institutions from most of the other countries.

The maximum contents of *Legionella* in A-samples and in B-samples in this study were much lower (five to six times lower) compared to the Danish care home study, even though the water temperatures were slightly lower in this study than in the care home study [9]. *Legionella* preventive procedures in a few care homes, such as the use of thermal shock up to 70 °C in water tanks once a week and regular cleaning procedures, resulted in an overall reduction in mean contents of *Legionella* of 70% after 2 months [7]. As almost all the schools in this study use such *Legionella*-reducing procedures, the general lower maximum contents of cultivable *Legionella* in the school hot water systems might be due to this prevention program. As discussed elsewhere, the effect of such procedures might stagnate after a while, indicating that *Legionella* reduction procedures might help reduce the maximum contents of *Legionella* in hot water systems. Still, these procedures are insufficient to remove *Legionella* from large hot water systems completely—over time.

The presence of *L. anisa* in some of the cultivated samples indicated that some of the schools have low water temperatures daily, as this species of *Legionella* is usually not found in water systems with high water temperatures [38,39]. The lack of *L. pneumophila* serogroup 1 in any of the samples further indicate, that the water temperatures are not very high in the investigated water systems, as this type often is selected if temperatures are high in water systems [38]. This is concerning because the hot water systems in all the investigated schools use BMS long-distance heat steering of the water systems, which should ensure that the water temperature is controlled and stays high enough to prevent the survival of *Legionella* at high levels.

However, low water temperatures might be a consequence for public institutions in the future because of a “sustainability demand” to keep energy use low when heating water. However, such procedures might increase the risk of the presence of high contents of *Legionella* in hot water systems (i.e., through protection from biofilm or through increasing growth of amoeba in the water system which is essential for growth of Legionella) [29]. If so, other treatments should be considered to avoid high levels of *Legionella*, such as softening the water (lowering calcium levels in the water) or using peripheral boosters in water systems [29].

As shown in Figure 1, temperatures above 50 °C were correlated with *Legionella* contents below 10,000 CFU/L. As shown in Figure 2, the tap water temperature must be at least 55 and 60 °C in central water tanks (Figure 3) to avoid *Legionella* contents above 1000 CFU/L in accordance with EU’s Drinking Water Directive [8]. These findings confirm that water temperatures above 50 °C but below 55 °C cannot guarantee low levels of *Legionella* in hot water systems [40]. A peripheral water temperature from taps of 55 °C seems sufficient to reduce the content of viable *Legionella* according to our results. Many Danish schools were built in the 1970s–1980s with an overcapacity in the water system compared to today’s needs for water consumption, with long and poorly insulated pipelines [7]. This may require a central water temperature of up to 60 °C or higher to ensure a high peripheral water temperature (above 55 °C), as indicated by Figure 2 and Figure 3. These characteristics also indicate a need to ensure a sufficient daily water flow and for regular use of all taps to prevent technical dead ends in the water systems.

A central water temperature of 60 °C in water tanks seems high to avoid high calcium precipitation in the water systems [11], as calcium precipitation promotes increase in biofilm growth, which protects the presence of *Legionella* [13]. Softening procedures, such as reducing calcium in domestic water or in local water systems, may, as mentioned elsewhere, be a solution if water temperatures must be raised to 60 °C or higher to avoid the presence of *Legionella* [29]. Another risk for schools with such high water temperatures is scalding among users, which would require local behavioural recommendations to avoid this issue [7].

## 5. Conclusions

*Legionella* was present in 97–100% of the hot water systems from the investigated schools in the two Danish municipalities from the capital region. The results showed that the schools in Municipality 1 had a significantly higher content of *Legionella* than those in Municipality 2. This is likely based on differences in water temperatures across the municipalities. It is recommended to consider the content and effectiveness of the *Legionella* reduction programmes, including regular heat shock, the cleaning of water tanks, and the control of temperatures by BMS in institutional water systems. If the programme is maintained, it is recommended to be revised to avoid high *Legionella* contents in many of the schools. Regular flow and draining procedures from all taps to avoid “technical blind ends” might be beneficial to include in local *Legionella*-reducing programmes, at least in some schools.

A test procedure of taking B-samples after 2 min combined with water temperature measurements after 3 min is suggested to ensure a more uniform and reliable test procedure for large water systems like those in schools. This study further determined that the peripheral water temperatures in taps should be increased to a minimum of 55 °C to maintain levels of culturable *Legionella* to a maximum of 1000 CFU/L, which constitutes the official limit value in EU countries for the presence of Legionella regardless of serotype (EU. 2021). This may require maintaining a central temperature of up to 60 °C or higher to ensure a high peripheral temperature (above 55 °C). Such high temperatures promote calcium precipitation in water systems that might increase the risk of Legionella growth. Therefore, a simultaneous reduction (softening) of the domestic or local water system could be a useful supplement to reduce Legionella exposure in Danish primary school hot water systems.

## Figures and Tables

**Figure 1 microorganisms-12-02074-f001:**
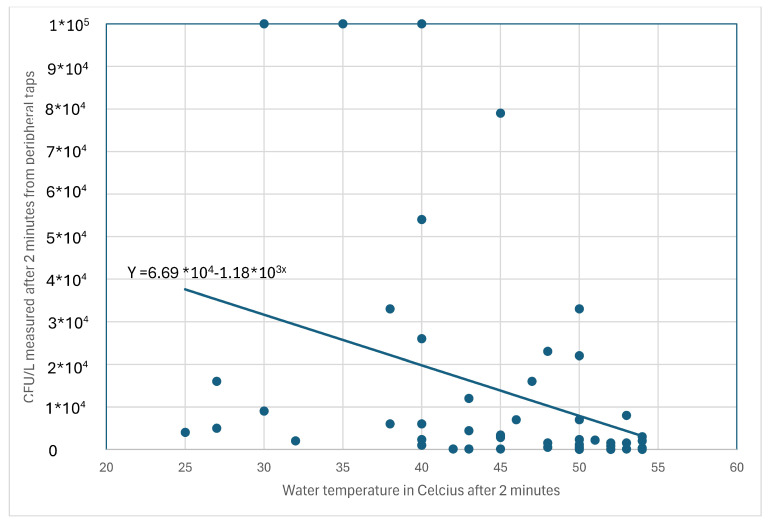
Legionella colony counts (CFU/L) as a function of water temperature, both measured after 2 min of flow from peripheral taps of hot water systems from 49 schools in the two Danish municipalities. In this figure is included a vertical reference line that marks 50 degrees Celsius and a horizontal reference line that mark 10,000 CFU/L.

**Figure 2 microorganisms-12-02074-f002:**
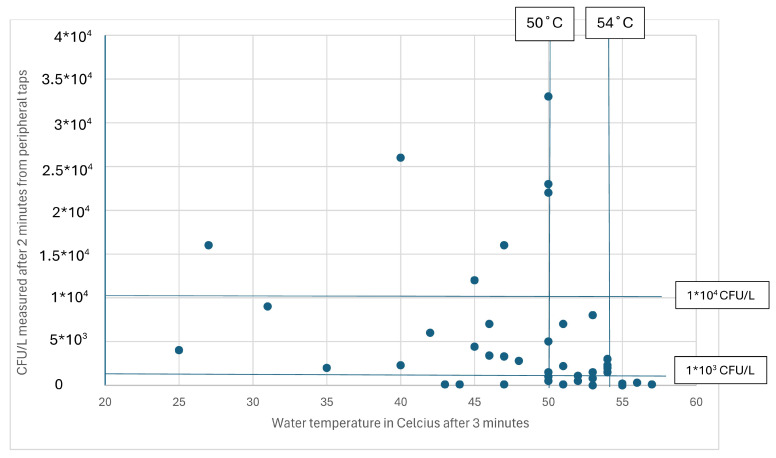
Legionella colony counts (CFU/L) related to water temperature from samples taken from peripheral taps of hot water systems in 49 schools in the two Danish municipalities. The water samples were collected after 2 min of flow, and the temperatures were measured after 3 min of flow from peripheral taps. In this figure are included vertical reference lines that mark 50 degrees Celsius and 54 degree Celsius and horizontal reference lines that mark 1000 CFU/L and 10,000 CFU/L.

**Figure 3 microorganisms-12-02074-f003:**
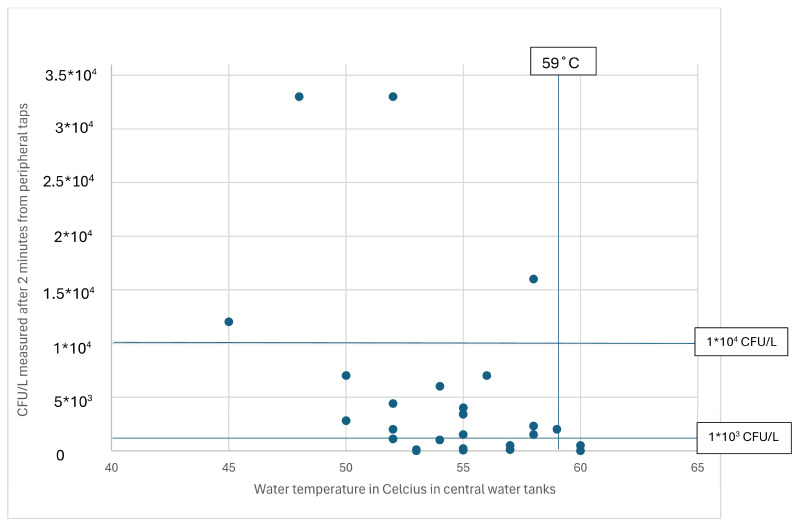
Legionella colony counts (CFU/L) to water temperature measurements in the central hot water tank in 49 schools in the two Danish municipalities. After 2 min of hot water flow from peripheral taps, water samples for CFU measurements were collected. In this figure are included vertical reference lines that mark 59 degrees Celsius and horizontal reference lines that mark 1000 CFU/L and 10,000 CFU/L.

**Figure 4 microorganisms-12-02074-f004:**
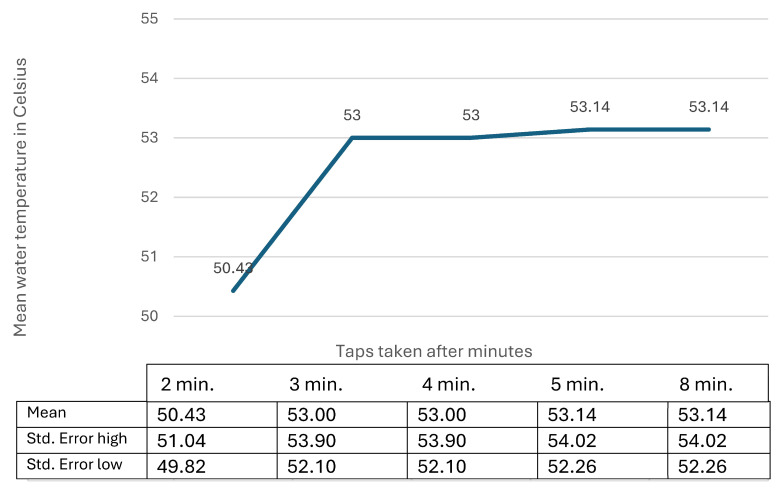
Mean water temperature (in Celsius) from hot water systems at seven schools from two Danish municipalities. Water temperatures were measured from taps after 2, 3, 4, 5, and 8 min of flow.

**Figure 5 microorganisms-12-02074-f005:**
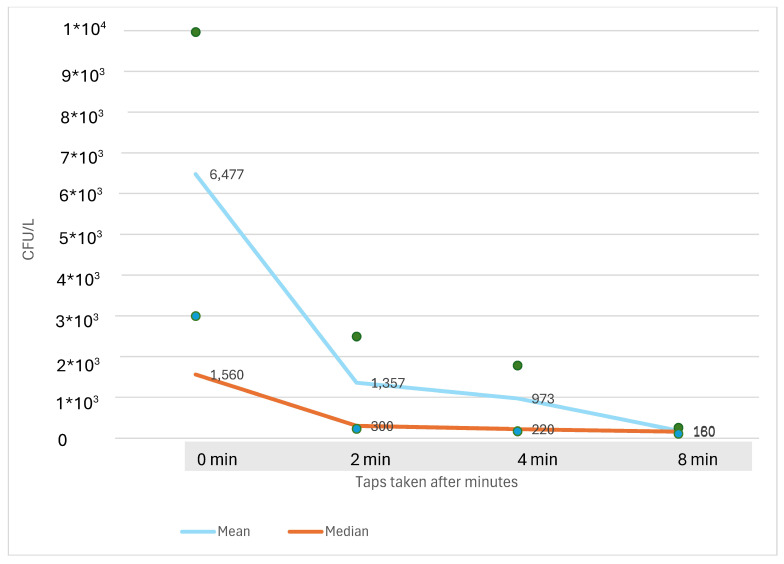
Mean and median contents of Legionella (CFU/L) from six schools from two Danish municipalities. Water samples were collected after 0, 2, 4, and 8 min. Standard error of mean around the mean line are marked by dots. The calculated mean CFU/L after 3 min of flow is marked on the mean line.

**Table 1 microorganisms-12-02074-t001:** Investigation of Legionella in hot water systems from schools in two Danish municipalities.

	Number of Investigated Schools in the Municipality	Average Number of Water Samples Per School	Proportion of Taps with >0 CFU/L in All Water Samples (A or B)	Proportion of Schools with ≥1000 CFU/L in Water Samples (A or B)	Proportion of Schools with ≥10,000 CFU/L in Water Samples (A or B)
Municipality 1 (n = 82)	41	2	97%	83%	63%
Municipality 2 (n = 16)	8	2	100%	87%	13%
Significance (*p*)			¤ *p* = 0.124 ns	¤ *p* = 0.742 ns	¤ *p* = 0.016 *

CFU/L: colony-forming units per litre (100 CFU/L is the lowest detectable limit for the presence of *Legionella*). *: significance level. ns: not significant. n: number of samples. ¤: *p*-value based on chi-square two-sided test.

**Table 2 microorganisms-12-02074-t002:** Investigation of Legionella contents in hot water systems from schools in two Danish municipalities.

	Highest Number of CFU/L in One Water Sample	Mean Number of CFU/L in Schools with *Legionella* (Median)	Mean Number of CFU/L in All Samples (Median)	Proportion of Schools with >0 CFU/L	Proportion of Schools with Mean 1000 ≤ 10,000 CFU/L	Proportion of Schools with Mean > 10,000 CFU/L
A-samples (n = 41) from schools in Municipality 1	100,000	31,268 (12,000)	30,505 (11,000)	97%	27%	51%
B-samples (n = 41) from schools in Municipality 1	100,000	17,300 (4700)	16,878 (4400)	97%	44%	32%
A-samples (n = 8) from schools in Municipality 2	21,000	4120 (1200)	4120 (1200)	100%	37%	13%
B-samples (n = 8) from schools in Municipality 2	3400	1850 (2150)	1393 (1250)	75%	50%	0%
Ratio and significance between B- and A-samples (B/A) from schools in Municipality 1	1.0	0.55 (0.33) ^ *p* = 0.064 ns # *p* = 0.005 ***	0.55 (0.40) ^ *p* = 0.057 ns # *p* = 0.005 ***	1.00 # *p* = 1 ns	1.62 # *p* = 0.655 ns	0.63 # *p* = 0.046 *
Ratio and significance between B- and A-samples (B/A) from schools in Municipality 2	0.16	0.45 (1.79) ^ *p* = 0.101 ns # *p* = 0.528 ns	0.34 (1.04) ^ *p* = 0.070 ns # *p* = 0.528 ns	0.75 # *p* = 0.157 ns	1.35 # *p* = 0.207 ns	0.0 # *p* = 0.317 ns
Ratio (M2/M1) and significance between mean A and between mean B samples (A/A & B/B) from schools in two municipalities	0.21 0.03	0.13 (A/A) 0.11 (B/B) ^ *p* = 0.005 *** ^ *p* = 0.041 *	0.14 (A/A) 0.08 (B/B) ^ *p* = 0.004 *** ^ *p* = 0.015 **	1.03 0.77 ¤ *p* = 1.000 ns ¤ *p* = 0.065 ns	1.37 1.14 ¤ *p* = 0.183 ns ¤ *p* = 0.202 ns	0.25 0 ¤ *p* = 0.059 ns ¤ *p* = 0.090 ns

M1 = Municipality 1. M2 = Municipality 2. CFU/L: colony-forming units per litre (detection limit is 100 CFU/L). *: 0.05 ≤ significance < 0.01; **: 0.01 ≤ significance < 0.001; ***: 0.001 ≤ significance. ^: Independent *t*-test. #: *p*-value for Wilcoxon signed-rank test. ¤: *p*-value based on chi-square two-sided test. Ns: not significant. N: number of samples.

**Table 3 microorganisms-12-02074-t003:** Relationship between water temperature categories and the Legionella contents in B-samples (stabile temperature) from hot water systems in schools from two Danish municipalities.

	Mean Water Temperature in Water Systems in Schools ≤ 50 °C (n)	Mean CFU/L in B-Samples from Schools with Water Temperature ≤ 50 °C (Median)	Mean Water Temperature in Water Systems in Schools > 50 °C (n)	Mean CFU/L in B-Samples from Schools with Water Temperature > 50 °C (Median)
Municipality 1 (n = 41)	41.5 (31)	21,694 (7000)	52.9 (10)	1950 (1500)
Municipality 2 (n = 8)	45.7 (7)	1577 (2000)	53.0 (1)	100 (100)
Significance (*p*) between Municipality 1/Municipality 2	# *p* = 0.426 ns	# *p* = 0.011 **	# *p* = 1.0 ns	# *p* = 0.027 *

CFU/L: colony-forming units per litre. *: 0.05 ≤ significance < 0.01; **: 0.01 ≤ significance < 0.001. Ns: not significant. #: *p*-value for independent *t*-test. N: number of samples.

## Data Availability

The data presented in this study are not available due to National ethical restrictions.

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
