# Peer review of "Legionella in Primary School Hot Water Systems from Two Municipalities in the Danish Capital Region"

_microorganisms, 2024, doi:10.3390/microorganisms12102074_

Round 1
Reviewer 1 Report
Comments and Suggestions for Authors
See the attached PDF

Author Response
-
- Abstract is directly started in illustrating results and conclusions which not clear, so it needs major revision.
Answer: Based on the reviewer’s feedback, the abstract needs to be revised to clearly introduce the study, its aims, methodology, and main findings, followed by clear conclusions. The abstract has been rewritten abstract based on the content of our study:
- Page 6/water samples
- This part does not present any information about the locations of schools or number of students in each location or any specific data for each location
Answer: Thank You for this feedback, which is correct. However, as per the agreement made with both the municipalities and the participating schools, their identities must remain confidential throughout the investigation. Therefore, we are unable to provide detailed location data or the number of students per location, as was also outlined in the cover letter. Nevertheless, it is important to note that the schools included in this study are similar in size, as most were constructed during the 1960s and 1970s. The water tank sizes and ages vary between schools, which is why these data were included in the analysis, as described in the manuscript. However, none of these factors had a statistically significant impact on Legionella concentrations.
- The methods of chemical analysis should be separated in different subtitles
Answer: Thank you for this statement. We have separated methods into different sections and made several revisions to this chapter for improved clarity. The methodology follows ISO Standard 11731, which we have now explicitly referenced in the text. This ensures that the procedures are well-described and adhere to the appropriate international standards for water sampling and Legionella analysis.
- It does not contain any tables or chart which may ease the understanding of collecting and analysis process
Answer Thank you for your comment. You are correct that the manuscript does not contain tables or charts specifically illustrating the collection and analysis process. However, the primary objective of this study is not to develop new laboratory guidelines for Legionella testing, but rather to present and analyze the results obtained using standard procedures as outlined in ISO Standard 11731.
- P 8/197 There is not a subtitle in any scientific paper could be named as (Figures and tables), its discussion should be merged in results and discussion sections!
Answer Thank you for your observation. We appreciate your input regarding the section title and have taken it into consideration. However, the text in this section serves primarily as a methodological explanation, providing context and linking the purpose of the figures and tables to the results. For this reason, we believe it is not appropriate to merge it into the Results section. We have decided to retain this section as it provides clarity and enhances the understanding of the results presented in the subsequent sections.
- Results/discussions are very confusing and not integrated as two sections, it should be merged in clear single sections
Answer: Thank you for your comment. We believe that merging the Results and Discussion sections may not comply with the journal’s criteria, which require separate sections titled "Results" and "Discussion". In the Results section, we present only the findings, which are structured according to the objectives outlined at the end of the Introduction and aligned with the Tables and Figures. The Discussion section starts by briefly summarizing the main results, followed by a discussion of potential biases and methodological considerations. We then compare our findings with those of other studies and conclude by discussing the implications of our results in relation to the main objective: addressing the new EU limits for Legionella in water systems. This includes suggestions for reducing Legionella levels in school hot water systems to comply with the new regulations. In this section, we integrate the topics from the Introduction with the study’s findings.
- All figure and tables are low quality, poor format and not well described in its captions
Answer. Thank You for that statement, we will improve those – especially avoid line breaks as is the case in some of the existing tables.

Reviewer 2 Report
Comments and Suggestions for Authors
This paper from Nielson and colleagues reports the results of a study of Legionella bacteria in the hot water of grade school in 2 districts in Denmark. They did quantitative cultures of water from taps collected immediately and after flushing the water for 2 minutes to eliminate stagnant water in the pipes. They also measured the temperature of all the collected water samples and in the hot water tanks. They did correlations between CFU/mL and water temperature. Essentially all the schools\s had Legionella in their systems. There were no reports of legionnaires disease from these school districts so I assume this was a convenience sample of schools.
The manuscript is well written, and the results are presented clearly.
I have only a few comments and suggestions.
1. This study was done in the summer. Is it possible that in the winter the water temperatures would be lower than in the summer?
2. Were there screens on the faucets of the sinks?
3. Line 91 replace elder with older.
4. L 285 the sentence is awkward. Did you mean “Colonies from the cultures from the seven schools that were analyzed were identified as either L. pneumophila serogroup 2-11 or L. anisa. No L. pneumophila 1 were identified.
5. The meaning of L 369 is not clear. I believe that sudy showed that the high water temperature temporarily prevented the growth of legionella, but with time at a lower temperature they were then able to cultivate legionella, and those organisms were tolerant to higher temperature, implying that hot water will not be itself ever sterilize a contaminated water system.
Comments on the Quality of English LanguageThe English writing is very good. I included a few suggestions for improvement in my comments to authors.
Author Response
- This paper from Nielson and colleagues reports the results of a study of Legionella bacteria in the hot water of grade school in 2 districts in Denmark. They did quantitative cultures of water from taps collected immediately and after flushing the water for 2 minutes to eliminate stagnant water in the pipes. They also measured the temperature of all the collected water samples and in the hot water tanks. They did correlations between CFU/mL and water temperature. Essentially all the schools\s had Legionella in their systems. There were no reports of legionnaires disease from these school districts so I assume this was a convenience sample of schools.
Answer: Thank you for your comment. We believe there may be a misunderstanding regarding the term "convenience sample." The schools included in this study were not selected based on convenience. In Municipality 1, approximately 50% of all schools were randomly invited to participate, and only one school declined. This school is the only one from the randomly invited group that did not take part in the study. In Municipality 2, 80% of the schools participated. We were unable to establish contact with the remaining schools, which are private institutions, unlike the public schools included in the study. Therefore, the study sample represents a substantial portion of the schools in both municipalities, selected through random invitation.
- The manuscript is well written, and the results are presented clearly.
I have only a few comments and suggestions.
- This study was done in the summer. Is it possible that in the winter the water temperatures would be lower than in the summer?
Answer: Thank you for that question. It is possible that the water temperature can show some variations between the seasons. Especially in the peripheral part of the water systems because a lot of the old schools have poor isolation around the peripheral tubes. How regularly the water is used (flow) is of cause also an important parameter regarding water temperatures and presence of circulation which some of the schools’ water systems are equipped with is another important point to be aware of.
Results from Danish Legionella studies through the last decades show a general increasing incidens of LD from early summer until fall. And from fall to winter periods a decrease in LD incidence. If high water temperatures are essential for lowering the Legionella contents in water systems and high contents of Legionella is essential for LD outbreaks, is it not likely with lower water temperatures during winter periods. But as described above, water temperature also depends on regularly use. The investigated schools do not decrease water temperature during ie. vacation periods. So, a relatively stable water temperature should be present in the water tanks all year. Decrease in water temperatures should therefore only be a possibility in the peripheral part of the water systems. This is why our suggestion - in this article - of measuring water temperatures after 3 minutes instead of the usually recommendations- taken after 2 minutes - might be a more realistic measurement. We will put in a note about that in the discussion chapter.
- Were there screens on the faucets of the sinks?
Answer: Thank you for that question. There were filters or perlators (water and air mixers) on most of the taps and our standard procedures included that these were removed before water samples were taken. None of the filter were though Legionella eg. POU filter types.
- Line 91 replace elder with older.
Answer: thank you for that correction
- L 285 the sentence is awkward. Did you mean “Colonies from the cultures from the seven schools that were analyzed were identified as either L. pneumophila serogroup 2-11 or L. anisa. No L. pneumophila 1 were identified.
Answer: Yes, thank you for that correction
- The meaning of L 369 is not clear. I believe that study showed that the high water temperature temporarily prevented the growth of legionella, but with time at a lower temperature they were then able to cultivate legionella, and those organisms were tolerant to higher temperature, implying that hot water will not be itself ever sterilize a contaminated water system.
Answer: Thank you for that opinion. We agree that especially short time raise in water temperatures is not enough to sterilize water systems – especially as Legionella bacteria are able to enter protozoans in a VBNC phase and later able to reenter the water system in an active mode when the conditions are right, and that thermo resistant types like L. pneumophila type 1 will do that first in higher water temperatures. We have included that statement in the discussion chapter.

Reviewer 3 Report
Comments and Suggestions for Authors
The article is a typical example of a case study, and this should be highlighted. In my opinion, it does not contain significant novelties, and in the form presented it is more of a mathematical compilation of data.
Of course, the problem of the incidence of Legionella bacteria and the diseases they cause is very important, but the article doesn't even provide precise information on what serotypes have been identified.
Even at the very beginning, the authors base only on Danish studies do not refer to other countries at least to compare the problem. It might be worthwhile to use, for example, the Surveillance Atlas of Infectious Diseases.
The methodology is described in such a way that in total it is not clear what was determined in what samples, only on the basis of the results it is easier to understand how the research was conducted.
The data presented in the graphs and tables are often duplicated - example figure 4 and table 4. Also tables 5 and figure 5 can be combined. The graphs are prepared carelessly - figure 5 lacks the title of the Y-axis, and on the X-axis, which relates to time, units of cfu/ l are inserted. Such duplication of results causes the reader to get lost in the values and begin to wonder what the authors intended to present.
The graphs for the number of microorganisms are more readable if the results are presented as 105 rather than a notation of 100,000.
line 237 It is puzzling how it is possible that there are no statistical differences between the mean of samples A and B. Wasn't the median taken into account here, rather than the mean?
Why does the detection limit of 10 CFU/L appear in the article once, and 100 CFU/L elsewhere? Were these conversions per liter? It should be explained how the detection threshold was calculated.
In conclusion, the article should be thoroughly rebuilt, consider what the main goal was, and present much more clearly the analysis performed.
Author Response
The article is a typical example of a case study, and this should be highlighted. In my opinion, it does not contain significant novelties, and in the form presented it is more of a mathematical compilation of data.
- Of course, the problem of the incidence of Legionella bacteria and the diseases they cause is very important, but the article doesn't even provide precise information on what serotypes have been identified.
Answer. Thank you for your question. Since the new EU Water Directive sets a Legionella limit of 1,000 CFU/L, regardless of serogroup or type, we believe it is more critical to assess Legionella concentrations across many schools to determine how many exceed this limit, rather than focusing on identifying specific serogroups in fewer systems. However, we did observe that L. pneumophila serogroup 1 was not present, while L. anisa was detected. This supports our observation in Table 3 that the water temperature is too low, as L. pneumophila serogroup 1 typically thrives in systems with higher temperatures, whereas L. anisa is more commonly found in cooler systems. This has now been included in the Discussion section.
- Even at the very beginning, the authors base only on Danish studies do not refer to other countries at least to compare the problem. It might be worthwhile to use, for example, the Surveillance Atlas of Infectious Diseases.
Answer. Thank you for your comment. This study focuses on investigating Legionella concentrations in Danish primary school water systems and does not include data on LD (Legionnaires' Disease) cases among pupils, or more relevantly, Pontiac Fever cases among youth. Since Pontiac Fever symptoms are often mistaken for influenza, such cases are rarely tested and registered, which likely leads to underreporting. A study investigating disease cases would require a completely different setup compared to our current investigation, which is focused solely on the presence of Legionella in the water systems.
In the discussion section – the lines 409-415 is included a comparison between this study’s results and results from institutions in other countries.
- The methodology is described in such a way that in total it is not clear what was determined in what samples, only on the basis of the results it is easier to understand how the research was conducted.
Thank You for this view. We have changed the method description regarding the 7 schools as we agree with reviewer that this description was too unclear. We have included a more in-depth explanation where a distinction is made between the standard tests taken after 0 and 2 minutes on all the investigated schools - and the special tests where several water samples and temperature measurements are taken at 7 schools. The text describing the analytic procedures is also slightly changed (see answers below).
- The data presented in the graphs and tables are often duplicated - example figure 4 and table 4. Also tables 5 and figure 5 can be combined. The graphs are prepared carelessly - figure 5 lacks the title of the Y-axis, and on the X-axis, which relates to time, units of cfu/ l are inserted. Such duplication of results causes the reader to get lost in the values and begin to wonder what the authors intended to present.
Answer: Thank you for this, there are some missing information in the figures and tables. Some of them based on transformation between word and pdf. That will be corrected. Do to reviewers suggestion we combine figures/tables 4 & 5 – meaning that tables 4 & 5 is removed as data is already included in the figures 4 & 5.
- The graphs for the number of microorganisms are more readable if the results are presented as 105 rather than a notation of 100,000.
Answer We will make a new table without line breaks.
- line 237 It is puzzling how it is possible that there are no statistical differences between the mean of samples A and B. Wasn't the median taken into account here, rather than the mean?
Answer Thank you for your observation. We understand that the lack of statistical difference between the mean values of samples may seem puzzling. However, this result is due to the variation in measurements combined with the chosen statistical method, which shows a strong tendency but does not reach statistical significance. To address this, we also report the significance between the median values to highlight the trend more clearly. The difference between the median and mean values often reflects large variations in the data.
- Why does the detection limit of 10 CFU/L appear in the article once, and 100 CFU/L elsewhere? Were these conversions per liter? It should be explained how the detection threshold was calculated.
Answer: Thank You for this statement. We now mention ISO standard 11731 in the method section to improve that part. We are sorry that the description is a bit short and unclear without this information. The limit of detection by the culture method is 10-100 CFU/L from filtering water samples with low concentrations of Legionella depending on the recovery. For samples with high concentration of Legionella the detection limit is 100 CFU/L from direct plating. As 100% recovery eg. depends on the sero-types the overall lower limit of content is set to 100 CFU/L. We change the lower limit to 100 CFU/L in the method section.
- In conclusion, the article should be thoroughly rebuilt, consider what the main goal was, and present much more clearly the analysis performed.
Answer: Thank you for this view. We agree that the purpose with comparing the school measurements with the EU standard of 1000 CFU/L needs to be specified in the purpose of this article. We hope that our answers and corrections are sufficient to clear the purpose and results of this article.

Round 2
Reviewer 1 Report
Comments and Suggestions for Authors
Thanks no more comments
Author Response
Thank You for your comments.
Reviewer 3 Report
Comments and Suggestions for Authors
The article reads much better after the corrections. However, in the version made available to me, the abstract has been removed and there is no new one entered. Is this some kind of mistake?
In addition, the graphs still show CFU records given as, for example, 10000 and not 105. And yet in microbiology, when declining in abundance, one looks primarily at the order of magnitude and reduction by a specific number of logarithms.
Although the article is interesting I still maintain that it is more of a case study of Legionella infections occurring in the country. Nevertheless, the amount of work done, including the number of samples taken, is very large, reflecting the actual state of Legionella prevalence in Danish schools.
Author Response
Thank You for your comments. We have changes the changed The axes notations in our figures as suggested in this comment:
‘In addition, the graphs still show CFU records given as, for example, 10000 and not 105. And yet in microbiology, when declining in abundance, one looks primarily at the order of magnitude and reduction by a specific number of logarithms’